# Effects of restricting social media usage on wellbeing and performance: A randomized control trial among students

**Avinash Collis** [1]* , **Felix Eggers** [2]*

1 McCombs School of Business, The University of Texas, Austin, Texas, United States of America,
2 Copenhagen Business School, Frederiksberg, Copenhagen, Denmark

☯ These authors contributed equally to this work.
* avinash.collis@mccombs.utexas.edu (AC); fe.marktg@cbs.dk (FE)

## Abstract

Recent research has shown that social media services create large consumer surplus. Despite their positive impact on economic welfare, concerns are raised about the negative association between social media usage and well-being or performance. However, causal empirical evidence is still scarce. To address this research gap, we conduct a randomized controlled trial among students in which we track participants' daily digital activities over the course of three quarters of an academic year. In the experiment, we randomly allocate half of the sample to a treatment condition in which social media usage (Facebook, Instagram, and Snapchat) is restricted to a maximum of 10 minutes per day. We find that participants in the treatment group substitute social media for instant messaging and do not decrease their total time spent on digital devices. Contrary to findings from previous correlational studies, we do not find any significant impact of social media usage as it was defined in our study on well-being and academic success. Our results also suggest that antitrust authorities should consider instant messaging and social media services as direct competitors before approving acquisitions.

## Introduction

Social media increasingly plays an important role in our daily lives. Ever since the launch of major modern social media platforms such as Facebook, users have adopted them at an explosive pace and adoption continues to increase to this day. Almost 3 billion users are active monthly on Facebook in 2021 [1]. This corresponds to over a third of the global population. The average adult spends over 45 minutes every day on social media platforms [2].

Given this rapid adoption and usage of social media platforms, it is essential to study the impact of social media on the well-being of users. Brynjolfsson et al. [3] find that digital technologies, including social media, generate a large amount of consumer surplus. More specifically, they conduct incentive compatible choice experiments to measure the consumer surplus generated by Facebook and find that the median US Facebook user obtains around $48/month of value from using Facebook in 2017 as measured from their willingness to accept to give up

**Data Availability Statement:** Raw data cannot be shared publicly since it is protected under GDPR. The data was collected at the University of Groningen and contains sensitive data including all online activities of students and their grades. The

IRB of the University of Groningen did not give the authors permission to share raw data. Please contact privacy@rug.nl if you have any questions regarding this. If you receive approval from the privacy office at the University of Groningen (privacy@rug.nl), we will share the raw data with you. Minimal data used to replicate the study's main results are provided at: https://osf.io/jfqp3/ (DOI 10.17605/OSF.IO/JFQP3).

**Funding:** A.C. received funding from the MIT Initiative on the Digital Economy for this research (https://ide.mit.edu/). There is no associated grant number. The funder had no role in study design, data collection and analysis, decision to publish, or preparation of the manuscript. F.E. received funding from the Behavioral Research Lab at the Faculty of Economics and Business at the University of Groningen for this research (https://www.rug.nl/feb/?lang=en). There is no associated grant number. The funder had no role in study design, data collection and analysis, decision to publish, or preparation of the manuscript.

**Competing interests:** The authors have declared that no competing interests exist.

access to Facebook for a month. They also conduct a similar experiment with students at a large European university and find that the median student in their sample obtains €97/month of value from using Facebook.

While Facebook and other social media services seem to generate a large amount of consumer surplus and contribute towards the economic well-being of their users, questions are raised about the negative externalities generated by social media. There is an active debate in media and academic research about the impact of social media on subjective well-being (including happiness and life satisfaction) and productivity. Current empirical results are ambiguous. Across different studies, correlational evidence points towards a positive, neutral (null results) and negative relationship between social media use and well-being (see Haidt [4] for a comprehensive literature review of social media use and mental health). However, most of this evidence suffers from issues related to reverse causality [5] and inaccurate measures of self-reported social media use [6]. Rigorous causal evidence on long term impacts of social media use on well-being is lacking.

Concerns are also raised in the field of Education policy on the impact of screen time (including social media use) on academic performance of students. Critics contend that social media use on smartphones distracts students from focusing in classes and affects their grades. Motivated by these concerns, the French education ministry banned smartphones in schools from first through ninth grades [7]. The American Academy of Pediatrics also recommends parents to limit the time spent by children and adolescents on social media so that they have enough time left to study [8]. However, a rigorous analysis of the data used in previous correlational studies that were used as evidence to support these policies suggests that the effects of social media use and screen time on adolescent well-being are too small to warrant policy changes [9].

Given these widespread concerns and conflicting correlational evidence on the impact of social media on well-being, it is necessary to obtain causal evidence in a timely manner before policies are implemented hastily. We seek to fill this research gap by conducting a randomized controlled trial to measure the causal long-term impact of social media use on academic performance and well-being. Specifically, the research question we aim to answer is: How does restricting social media use affect a) well-being, b) academic performance, and c) other digital activities?

We recruit students at a large European university to be part of our study over the course of three academic terms (quarters). The subjects install a software (RescueTime) on their personal computers and mobile devices. This software tracks all of the digital activities of the subjects during the entire duration of the study period. The first term serves as the baseline period. In the second term, subjects are randomized into treatment and control groups and the treatment group has social media use (Facebook, Instagram and Snapchat) restricted to a maximum of 10 minutes per day across all devices. We then measure the post-treatment effects in the third term.

We observe the entire space of digital activities performed by our subjects that covers online and also offline activities on their devices, including activities related to learning (such as writing in Microsoft Word or reading a PDF). Our social media use metrics are computed based on the actual time spent on social media and are not based on self-reported metrics of time spent, which is predominantly used in the existing literature. In addition to the digital activities, we obtain objective metrics of performance (grades) and subjective well-being scores solicited through surveys.

Contrary to results from previous studies using observational data, we do not find evidence that social media causes a positive or negative impact on well-being (including life satisfaction and mental health). Moreover, we also do not find any evidence that social media usage

impacts academic success. However, we find significant substitution effects. Specifically, we see that participants in the treatment group substituted their use of social media services for instant messaging apps (e.g. WhatsApp). In total, these participants do not spend less time on their digital devices (computers and mobile phones) as those in the control group.

Our paper makes three main contributions. First, we test the popular media narrative portraying social media as the villain responsible for negatively affecting well-being of society. We do not find any evidence supporting this hypothesis. Second, educators and parents are increasingly concerned about the impact of digital distractions on academic performance and are restricting the online activities of students (for example through parental control software or by taking away their devices). While previous evidence seems to suggest that device usage in class might negatively affect academic performance, our results show that restricting social media usage from the lives of students (inside and outside class) might not have the intended effect. Finally, our paper is one of the first to provide evidence of substitutability between social media and instant messaging apps. This has major implications for antitrust authorities analyzing the market power of major social media platforms such as Facebook which owns Instagram (another social media service) and WhatsApp (instant messaging service).

The paper proceeds as follows. In the next section, we provide a brief review of existing literature on the impact of social media use on well-being and academic performance. In the following section, we describe the design of our experiment and data collected over the course of the study. We then show the main results and conclude with a discussion of the limitations of this study and directions for future research.

## Related literature

The impact of the internet in general, and social media in particular, on well-being has attracted the attention of a number of researchers in the fields of psychology, epidemiology and human-computer interaction (HCI) over the past two decades. Most of this literature uses self-reported metrics of technology use and provides cross-sectional correlational evidence. Kraut and Burke [10] provide a review of this literature and express skepticism regarding cross-sectional and survey-based studies due to the presence of several confounding factors. Moreover, correlational studies might suffer from an abundance of researcher degrees of freedom and the file drawer problem such that only significant results are published, inevitably leading to the implication that social media usage either has a positive or negative effect. However, a null result or insignificant findings regarding social media usage might be a plausible outcome.

Orben and Przybylski [9] rigorously analyze popular large scale social datasets (n = 350k) used in previous correlational studies studying the impact of technology use on well-being by conducting a specification curve analysis of the data. This analysis involves running all possible analytical models using various combinations of the covariates. Instead of selective reporting, results from all of these analyses are reported. They find a small negative association between digital technology use and adolescent well-being. However this effect is economically insignificant explaining at most 0.4% of the variation in well-being. For comparison, the authors show that seemingly neutral activities such as eating potatoes have the same negative association with well-being as technology use. Given these concerns with correlational analyses involving cross sectional data, Kraut and Burke [10] call for experimental evidence paired with tracking data to provide reliable evidence on the relationship between internet use and well-being.

The subset of literature focusing on the association between social media use and well-being has found a wide range of effects (negative, mixed, positive and null). Using a longitudinal survey, Shakya and Christakis [11] found a negative association between Facebook use and

well-being. In contrast, Burke et al. [12] find a positive association between directed communication on Facebook and social well-being due to subjects reporting improved feelings of social bonding and reduced loneliness. Similarly, Hobbs et al. [13] match Facebook profiles with public health records and find that being more socially integrated online (by accepting more Facebook friends) is associated with reduced risk of mortality. Burke and Kraut [14] find that targeted messages from strong ties is associated with positive improvements in well-being while viewing messages from friends broadcasted to all of their friends and receiving one-click feedbacks were not associated with any improvement in well-being. Kim and Shen [15] find that directed communication activities on social media platforms are positively associated with life satisfaction for older users.

Schemer et al. [16] and Johannes et al. [17] do not find any substantial relationship between social media use and subjective well-being. Liu et al. [18] conduct a meta-analysis of 124 studies and find that the association between social media use and well-being depends on the type of social media use. This association is positive for social interactions and negative for content consumption. Huang [19] also conducted a meta-analysis of 61 studies and documented a weak correlation between time spent on social media and negative indicators including depression and loneliness. For a comprehensive list of all studies studying social media use and well-being, see literature reviews by Appel et al. [20], Meier and Reinecke [21], Dienlin and Johannes [22], Masur [23] and Orben [24]. Cheng et al. [5] combine a survey of Facebook users with their Facebook activities and find that subjects reporting problematic use of Facebook were also going through a major life event such as a breakup. This shows that confounding variables could be a major concern in previous studies associating social media use and well-being.

Orben et al. [6] use a large-scale longitudinal dataset and conduct a specification curve analysis to rigorously analyze the relationship between adolescent social media use and well-being. Most of the analyses report tiny, trivial and insignificant results. Moreover, they provide evidence for reverse causality showing that social media use predicts well-being in the future and vice versa. Similarly, Sewall et al. [25] also find little to no evidence that changes in social media or smart phone use predict psychological distress.

Another major concern related to existing studies is the use of self-reported usage data. Survey respondents are typically asked to report the average time they spend on the internet, social media and digital devices. Several papers show that self-reported measures of technology use (including social media usage) are poorly correlated with actual usage and contain systematic patterns of misreporting [26–29].

Given this inconclusive evidence and lack of objective technology use data in existing literature, it is essential to obtain reliable causal evidence in a timely manner to inform policy makers. We aim to resolve this gap by obtaining evidence through a randomized controlled trial and using objective technology use metrics tracked by a software installed on the digital devices of our experimental subjects. In terms of outcome variables, we track measures of subjective well-being (life satisfaction and mental health) and performance (grades and number of credit points) over the duration of three quarters of an academic year (8 months) with the actual treatment lasting 2.5 months.

There is a related stream of literature using experiments to study the relationship between social media or computer usage and well-being or performance. Verduyn et al. [30] conduct a lab experiment where subjects are primed to passively use Facebook for 10 minutes and find that passive use is associated with a decline in subjective well-being. However, it is not straightforward if results from a 10-minute treatment can be generalized to long term effects.

Marotta and Acquisti [31] conduct an experiment with workers recruited from Amazon mechanical turk and offer productivity enhancing tools to subjects. One of the treatment

groups has popular social media sites blocked during work hours. They find that workers in this group completed more tasks and increased their earnings. Carter et al. [32] conduct a randomized controlled trial in a US university where classes in the treatment group prohibited the use of computers in the class. They find that average exam scores were higher in the treatment group compared to the control group classes where students were allowed to use their computers. Using causal inference methods on observational data, Belo et al. [33] and Beland and Murphy [34] study the impact of broadband access and banning mobile phones in schools respectively on academic performance and also find evidence suggesting that digital distractions during class reduce academic performance. Taken together, evidence seems to suggest that digital device use in class or at work is harmful for student or worker performance. However, the overall causal impact of social media usage in life (inside and outside class or at work) on performance and well-being still remains an open question. Our study complements this research by analyzing the overall long-term impact of social media on well-being and performance as the subjects in our treatment group has restricted use of social media throughout their day for a long period of time.

A study closely related to our research is the experiment conducted by Allcott et al. [35]. They conducted a randomized controlled trial of Facebook users where subjects in the treatment group had to deactivate their Facebook account for 1 month. They find that this treatment reduced total online activity including other social media and this reduction persists after the end of the experiment. However, they use self-reported metrics of usage of online activities which are weakly correlated to objective usage metrics according to previous research. They measure 11 different metrics of subjective well-being and find that deactivating Facebook led to increase in subjective well-being for 4 out of the 11 metrics. Other related experiments include Brailovskaia et al. [36] (two weeks intervention, self-reported Facebook usage measure), Hall et al. [37] (one to four weeks intervention, self-reported Facebook usage measure), Hinsch and Sheldon [38] (one week intervention, self-reported Facebook usage measure), Tromholt [39] (one week intervention, self-reported Facebook usage measure), Mosquera et al. [40] (one week intervention), Hanley et al. [41] (one week intervention) and Vanman et al. [42] (five days intervention). Overall, the magnitude of the effects is small and it is not clear if these effects would have persisted for a treatment of longer duration. For a longer treatment duration, subjects could learn to live in a world without Facebook by discovering alternative substitutes providing similar use cases and their subjective well-being scores could go back to pre-experiment levels.

## Methods

### Experimental procedure

We recruited students in the faculty of economics and business of a large European university to take part in an academic study. The study received approval from the Institutional Review Board of the university. We used a flyer to invite students in lectures and from the pool of participants of the behavioral research lab of the university. The flyer informed students about the subject of the study, the required activities, the reward, and about measures to protect the participants' privacy. Specifically, we let the students know that the study required to install a software (RescueTime) on their computers and mobile devices that keeps track of their digital activities and that allows them to analyze how much time they are spending on various categories of activities. We also stated that the study tracks their academic performance and well-being. Moreover, we informed the students that, in order to qualify for the reward, they need to keep the software running during the time of the study and to take part in four online surveys; one at the beginning of the study and one after each quarter. In addition to getting the

software for free, we offered students €20 and a one out of 100 chance to win €1,000 if they take part until the end of the study.

The sign-up link forwarded interested students to a registration form that provided a more detailed privacy statement (about aim and principles of processing personal data, the types of data used, the limited recipients of the data, the storage period, and the students' rights), informed consent, and asked students for their student email address, basic information about their studies (program, year), and the number and type of computers and mobile devices. The registered students were then invited to the study according to the experimental design detailed below.

## Experimental design

The recruitment of students took place in the first quarter of the academic year 2018/19. We scheduled the experiment to run for the remaining three academic quarters. We will refer to these three terms as block 1, 2, and 3 of the study (which are quarter 2, 3, and 4 of the academic year). Each block consists of seven weeks of teaching and two examination weeks. The specific timeline was:

- Block 1: from mid-November to end of January, with holidays from December 24 to January 3.

- Block 2: from February to mid-April.

- Block 3: from mid-April to end of June, with holidays from April 19 to 24.

We used the first block to establish a baseline of the students' digital activities. In block 2, we randomly assigned participants to one of two conditions: a control group without specific instructions and a treatment group that received an incentive to use social media as little as possible. Specifically, we instructed them to use Facebook, Instagram and Snapchat for a maximum of 10 minutes per day. We did not block these services completely because not having access to social media at all might have a negative effect on students, e.g., if they use it to exchange important information about their studies. The 10-minute limit enables students to still access relevant information while not allowing them to waste a longer period of time. This is consistent with the Goldilocks hypothesis according to which moderate digital use may be advantageous compared to no use or overuse [43]. The software would inform students in the treatment group when they reached the limit and automatically block Facebook, Instagram and Snapchat afterwards. Students could disable this feature if they needed to use these services for longer. We informed students that we gave away another €1,000 among all students who achieved to stay under the 10-minute limit throughout block 2. Block 3 served to assess post-treatment effects.

We focus on Facebook, Instagram, and Snapchat in our experiment because they were the most popular social media services according to our measurement in block 1. As a caveat, the distinction of these services from others such as instant messaging can be debated. We return to this issue in the discussion.

We invited students to four surveys in total. We have sent the first survey in the first week of block 1. This survey asked students to give informed consent and, after referring them to the privacy statement, their agreement to use their academic grades for the purpose of the study. Moreover, we asked them about basic demographic information, their study program, and additional work activities next to their studies. Moreover, we provided measures of subjective well-being (see specific measures below). Upon completion of this first survey, we gave students the installation and registration instructions for the tracking software and asked them

to keep this software running henceforth on all their computers and mobile devices. While the software was supported by Windows, OS X, and Android devices, it was not compatible with iOS devices (iPhone or iPad). In order to make sure that students with iOS devices complied to the 10 minute limit in the treatment condition, we informed them that we will ask them at a random time to hand in a screenshot of the Screen Time feature of iOS that reports similar information.

Surveys 2, 3, and 4 followed after each block and repeated the subjective well-being measures in order to track students' well-being over time. We gave students a one-week deadline to fill out each survey.

## Survey measures

As measures of subjective well-being we use the satisfaction with life scale (SWLS) [44] that consists of five items (In most ways my life is close to my ideal; The conditions of my life are excellent; I am satisfied with my life; So far I have gotten the important things I want in life; If I could live my life over, I would change almost nothing). These items are measured on a 7-point scale (1 "strongly disagree" to 7 "strongly agree"). SWLS is the most widely used scale to measure subjective well-being and is also used in previous studies studying social media use and well-being (e.g., [30,45]). In addition to SWLS, we also collected direct measures of happiness and life satisfaction through standard questions widely used in previous literature. Besides numerous other studies, the happiness question is used in the World values survey [46] and the life satisfaction question is used by Gallup [47] to calculate its well-being index. These questions are highly correlated with objective measures of well-being such as brain activity, emotional expressions and suicide rates as well as decision utility [48]. We obtain qualitatively similar results using these happiness and life satisfaction scores as we found using SWLS.

For measuring mental well-being, we adopted the shortened Warwick-Edinburgh Mental Well-being Scale (SWEMWBS) [49,50] with seven items (I've been feeling optimistic about the future; I've been feeling useful; I've been feeling relaxed; I've been dealing with problems well; I've been thinking clearly; I've been feeling close to other people; I've been able to make up my own mind about things). These items are assessed on a 5-point scale ranging from "None of the time" to "All of the time". The SWEMWBS is a popular scale to measure mental well-being and is used in previous studies studying technology use and mental well-being [43].

## Overview of data sources

Overall, our study makes use of three data sources: digital activities tracked by the software, self-reported measures via surveys, and academic grades from the educational administration. Table 1 shows an overview of these data types.

The software tracks users' activities on each device in 5-minute intervals and records how many seconds a user has actively used a specific program, app, or website in this interval, ranging from 1 to 300 seconds. Specifically, it records the user id, the name of the activity, the

**Table 1. Overview of data sources.**

| Type | Measure | Source | Data collection |
|---|---|---|---|
| Digital activities | Usage in number of seconds | Tracked by software | On each participant's device throughout the entire study |
| Subjective well-being | Rating scales | Self-reported in surveys | At the beginning of the study and after each teaching block |
| Academic grades | From 0 to 10, with <6 = failed, 6 = below standard, 7 = standard, >8 above standard; | Educational administration | Once at the end of the academic year |

system name (Windows, Mac OS, or Android), and a timestamp. Since we are specifically interested in social media activities of the three most used social network services Facebook, Instagram and Snapchat, we used lookup tables to classify activities accordingly. For example, Facebook usage could appear in the activities as "facebook.com", "fb.com", "messenger.com", "Facebook for Android", "Facebook for Windows", etc. We gathered this list of activities in block 1 and used each of these activities to count toward the 10-minute limit for the treatment group in block 2.

The European university at which this study took place uses a grading system that ranges from 0 to 10. Any grade below 6 represents a fail. A grade of 7 is most common and often referred to as "standard", a 6 as "below standard", and an 8 or higher as "above standard". The grade for a lecture typically consists of a combined grade of the final exam and assignments that have to be completed during the course.

## Sample

A total of 191 respondents completed the first survey. As is typical for longitudinal studies, some students dropped out over time such that 157 students completed survey 2, 144 survey 3, and 121 the final survey. The survey participation corresponds to the number of participants who reported digital activities using the software (see Table 2).

The following results will be based on the sample that recorded activities for at least 30 days in block 1 and 2 and completed surveys 1, 2, and 3. We will analyze the post-treatment data from block 3 and survey 4 separately. From the 134 students who recorded activities in block 1 and 2, we were able to match 122 from all data sources, i.e., twelve students did not answer (one of) the surveys or did not follow courses in at least one of the blocks.

Despite the dropouts, most importantly, there are no significant differences between the treatment and the control group, in terms of gender (Chi-squared test, p = 0.471), age (t-test, p = 0.961), mobile device operating system (Chi-squared test, p = 1.000), number of years studying at the university (t-test, p = 0.334) or whether students are working next to their studies in block 1 (Chi-squared test, p = 0.974) or block 2 (p = 0.594) (see Table 3 for details). There are also no significant differences between those who started the study and those who dropped out in terms of gender (Chi-squared test, p = 0.701), age (t-test, p = 0.113), mobile operation system (Chi-squared test, p = 0.975), of work status in block 1 (Chi-squared test, p = 0.109) or block 2 (p = 0.169). However, there is a significant difference between these samples regarding the study year (t-test, p = 0.027) such that those who dropped out are more likely to be Bachelor degree students than Master's students. One potential explanation is that Bachelor degree students are more likely to quit their studies and not have any courses or grades registered subsequently.

**Table 2. Number of participants.**

| Part | Number |
|---|---|
| Completed survey 1 | 191 |
| Used software in block 1 (calibration) | 149 |
| Completed survey 2 | 157 |
| Used software in block 2 (treatment) | 134 |
| Completed survey 3 | 144 |
| Used software in block 3 (post-treatment) | 125 |
| Completed survey 4 | 121 |
| Took courses in block 1, 2, and 3 | 158 |

**Table 3. Descriptive statistics of the sample.**

|  | Treatment | Control |
|---|---|---|
| Number of students | 60 | 62 |
| Gender: Female (vs. male) | 0.467 | 0.548 |
| Age (SD) | 22.1 (3.3) | 22.1 (3.1) |
| Mobile device operating system: Android (vs. iOS) | 0.500 | 0.500 |
| Studying in first to third year (vs. more than three years) | 0.667 | 0.629 |
| Working next to studying (block 1) | 0.400 | 0.419 |
| Working next to studying (block 2) | 0.500 | 0.435 |

For the subsequent analysis, we note that at 80% power and alpha = 5%, the minimum detectable difference with our sample size is 2.5 for SWLS (assuming a standard deviation of 5.0), 1.5 for SWEMWBS (standard deviation = 3.0), and 0.5 for academic grades (standard deviation of 1.0).

## Results

### Digital activities

**Social media usage.** On average, students tracked 223.7 minutes of digital activities per day across the entire study (SD = 115.1 minutes). Students who use an Android mobile device recorded significantly more activities (265.6 minutes; t-test, $p < 0.001$), compared to students with an iOS device (182.5 minutes) as iOS was not supported by the software. While our activity estimates are more accurate for Android users we expect the treatment condition to be equally effective for both of these segments because we informed participants to also inspect their iOS tracked activities (see above).

Fig 1 shows the total number of minutes tracked by day, averaged for students with Android mobile devices in the treatment (black dots) and control groups (white dots). We report the activities for users with Android devices because the tracking is more accurate as it captures activities on desktop/laptop computers and their mobile devices (figures corresponding to the overall sample are in the Supporting Information, S1 Fig). The solid vertical lines separate blocks 1, 2, and 3 and the dashed vertical lines indicate the start of the examination period. Overall, digital activities remain on a high level each day but are reduced during the winter holiday season and during the examination periods.

As a manipulation check, the bottom part of Fig 1 shows activities for social networking (Facebook, Instagram, and Snapchat combined) for the Android sample. The mean daily usage in minutes is 21.1 minutes (27.9 minutes) for users in the treatment (control) group in block 1, which is not significantly different (t-test, $p = 0.310$). The incentive to reduce social media activities was effective as students in the treatment condition significantly reduced their social media usage in block 2 compared to the control group (t-test, $p = 0.009$). The horizontal line represents the 10-minute limit imposed on the treatment group. The average usage per day is close to the limit in the treatment group with 8.1 minutes. Within the control group, the average daily usage of 24.2 minutes in block 2 is on the same level as in the first block (paired-sample t-test, $p = 0.245$).

Remarkably, although students in the treatment group significantly reduced their social media activities, their overall digital activities overall are not affected but, in fact, exceed those of the control group in block 2 (t-test, $p = 0.026$). This result indicates that students substituted or even overcompensated their social media usage with other activities.

## All digital activities

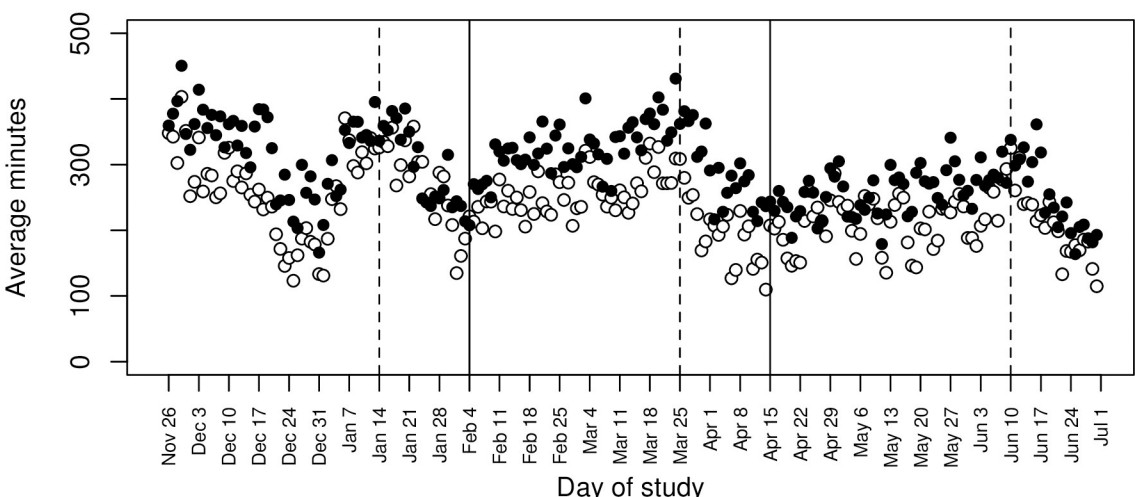

## Social media usage (Facebook, Instagram, Snapchat)

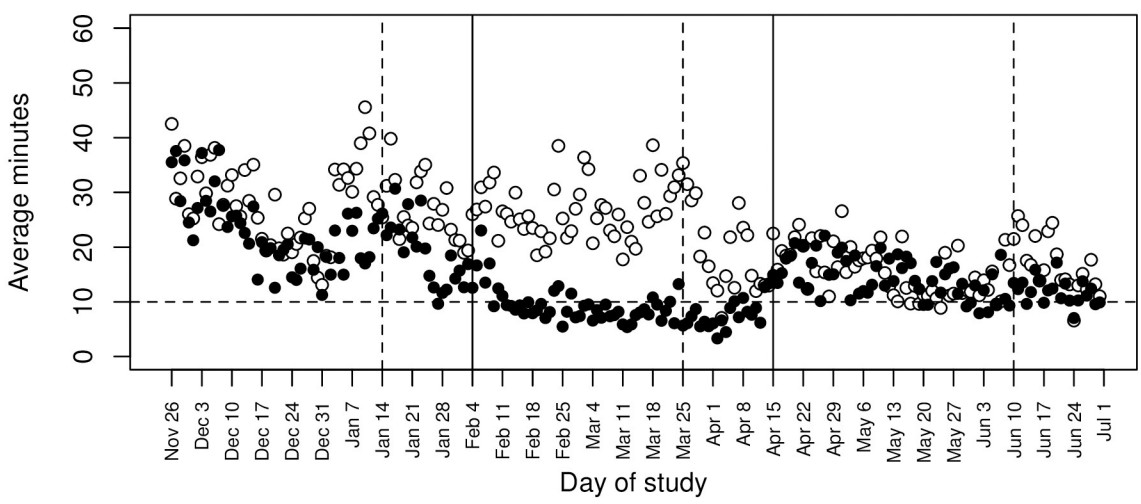

black = treatment, white = control group
solid vertical lines: start of a new teaching block
dashed vertical lines: start of the exam period

**Fig 1. All digital activities and social media usage over time (users with Android devices).**

*Substitution*. Fig 2A and 2B show the time series of activities of users with an Android device for the most used categories of services (we exclude the categories of general utilities, which holds mostly operating system activities, and uncategorized services for which there are no significant differences between the groups; activities for all users are in the Supporting Information, S2A and S2B Fig). We find significant substitution for social networking with instant messaging (p = 0.008). Accordingly, more students in the treatment condition used

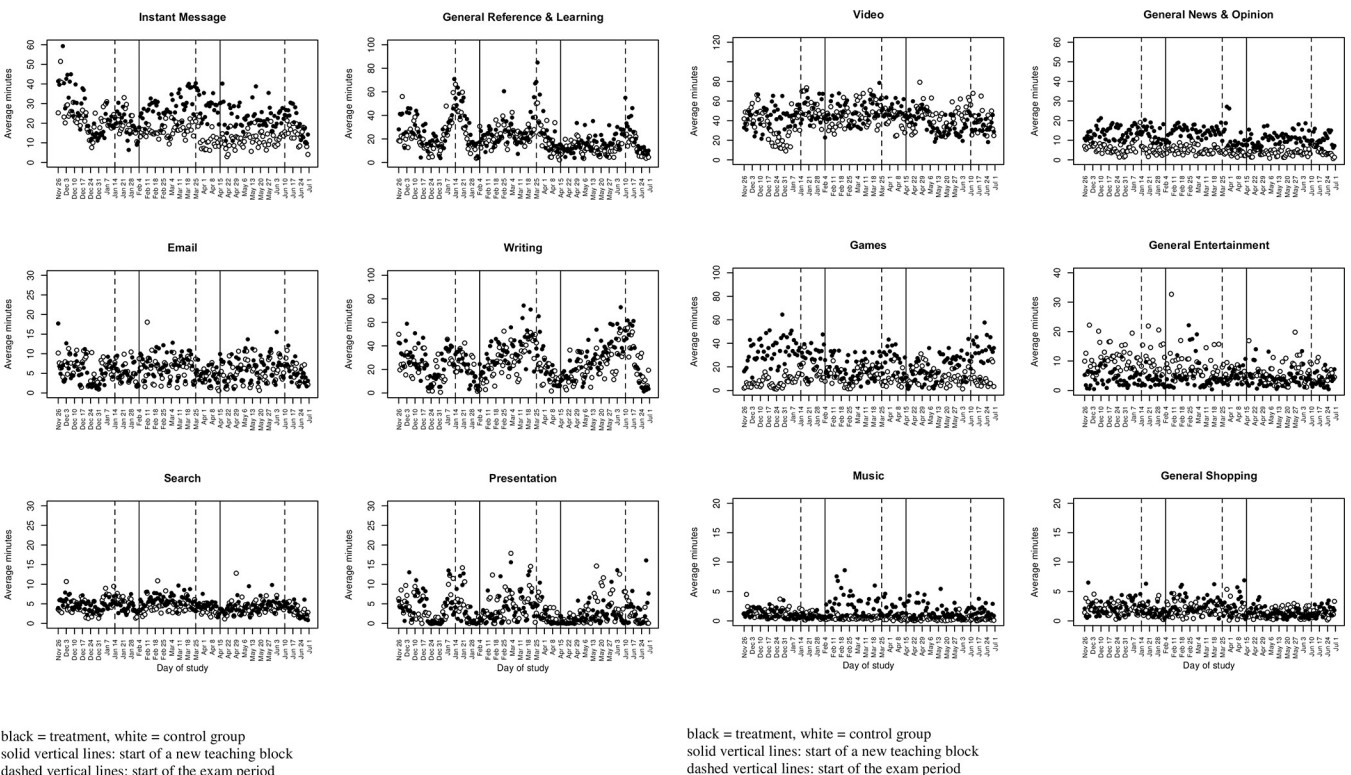

black = treatment, white = control group
solid vertical lines: start of a new teaching block
dashed vertical lines: start of the exam period

black = treatment, white = control group
solid vertical lines: start of a new teaching block
dashed vertical lines: start of the exam period

**Fig 2.** a. Tracked digital activities over time (users with Android devices). b. Tracked digital activities over time (users with Android devices).

instant messaging in block 2 when social media was restricted compared to block 1 and to the control group (difference-in-differences). The activities increased from an average daily use of 25.1 minutes in block 1 to 28.8 minutes in block 2 in the treatment group, while the usage decreased from 21.8 minutes to 15.2 minutes in the control group. Most activities (92.9%) in this category are related to WhatsApp.

We also find a significant increase in usage of music for Android users (t-test, p = 0.027) in the treatment group in block 2. However, average daily activities in this category are rather low (below five minutes) and the difference is mostly driven by two outliers who listen to music for more than 30 minutes each day on average. Other activities show plausible patterns, e.g., the reference and learning category (activities include the university intranet, PDF reader, Wikipedia, Mendeley, Google scholar, EBSCO, etc.) shows peaks before the exam period. However, these and other activities are not affected by reduced social media usage (an overview of significance tests comparing treatment and control groups in block 2 vs. block 1 is given in the Supporting Information, S1 Table).

**Subjective well-being.** For the subjective well-being measures, the SWLS and SWEMWBS items show high reliability (Cronbach's $\alpha$ being 0.84 and 0.76 respectively). For the analysis we calculate scores as the sum of their items (with SWEMWBS being transformed according to a defined conversion table). The students score averages (SD) in SWLS of 25.0 (5.5), 25.0, (5.4), 25.1 (5.3) in the three surveys at the beginning of the study and after block 1 and 2. That means they are between an "average" and "high" score of satisfaction. Treatment and control group are not significantly different at the beginning of block 1 (t-test, p = 0.182; survey 1), at the end of block 1 (t-test, p = 0.212; survey 2), or, most importantly, at the end of block 2 after the exposure to the treatment (t-test, p = 0.167; survey 3). The same implications hold for the

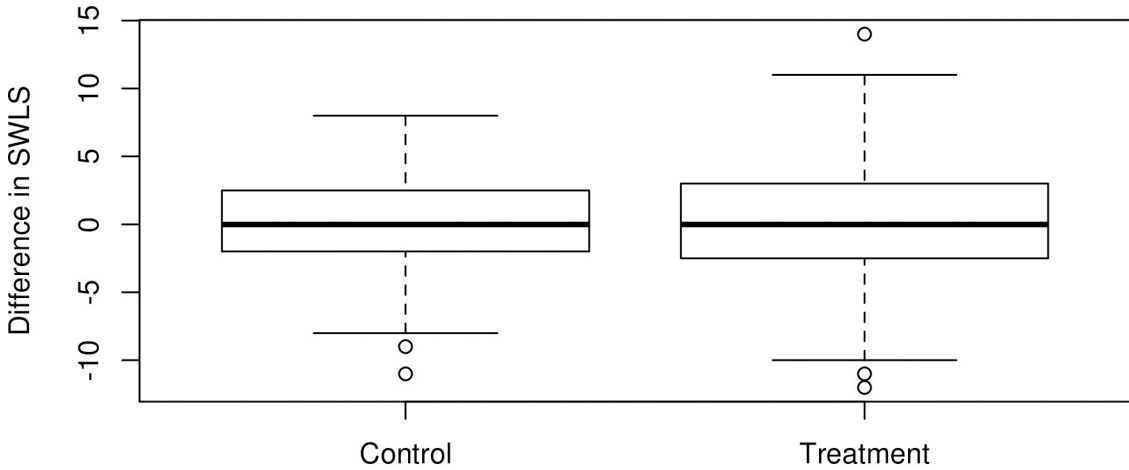

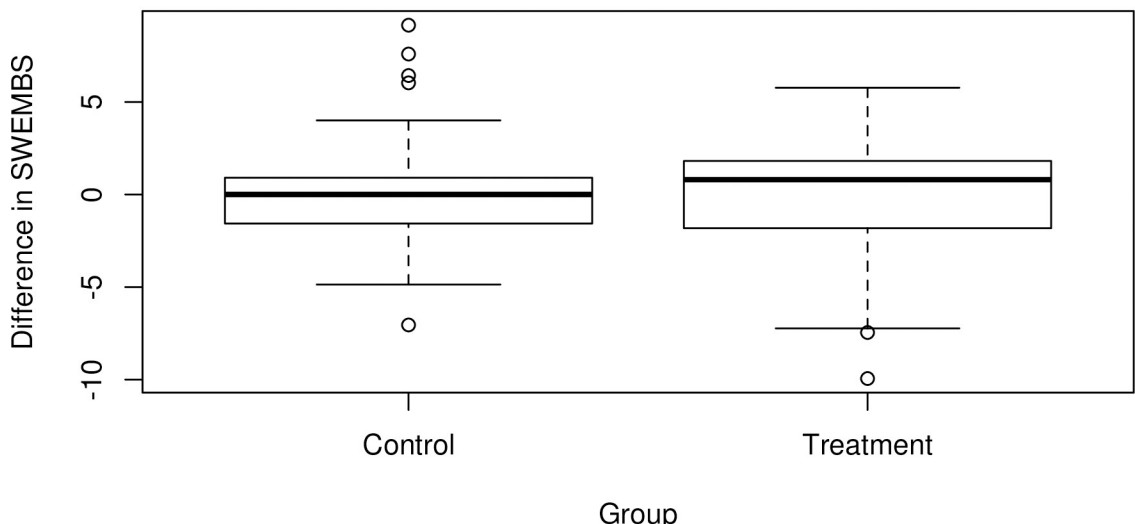

**Fig 3. Differences in subjective well-being measures.**

SWEMWBS scores that shows average scores (SD) of 22.8 (3.0), 22.5 (2.7), and 22.4 (3.3) in the three surveys (see S2 Table in the Supporting Information for details).

Fig 3 plots the differences between survey 3 and survey 2 (before and after the social media restriction) in terms of SWLS and SWEMBS. The distributions are centered on zero, illustrating the non-significant difference between the treatment and control group.

Table 4 shows correlations with the subjective well-being measures and the digital activities in block 1 (i.e., activities that are not affected by the treatment condition) for users with an Android device. We detail the correlations of the subjective well-being measures at the beginning of the first block (in survey 1) and at the end of the block (in survey 2) to study potential reverse causality, e.g., increased well-being leads so more/less social media activities or vice versa.

**Table 4. Correlations of subjective well-being measures and digital activities in block 1 (Android users).**

| | SWLS Survey 1 (start of block 1) | SWLS Survey 2 (end of block 1) | SWEMWBS Survey 1 (start of block 1) | SWEMWBS Survey 2 (end of block 1) |
|---|---|---|---|---|
| SWLS Survey 2 (end of block 1) | **0.80** | | | |
| SWEMWBS Survey 1 (beginning of block 1) | **0.76** | **0.63** | | |
| SWEMWBS Survey 2 (end of block 1) | **0.63** | **0.72** | **0.77** | |
| All digital activities | -0.06 | -0.11 | -0.15 | -0.05 |
| Social media usage | -0.03 | -0.02 | -0.13 | -0.02 |
| Social media usage low | **-0.31** | -0.23 | -0.08 | -0.13 |
| Social media usage medium | **0.26** | 0.19 | 0.14 | 0.15 |
| Social media usage high | -0.02 | -0.01 | -0.08 | -0.05 |
| General Reference & Learning | -0.13 | -0.14 | -0.20 | -0.06 |
| Instant Message | 0.18 | 0.24 | 0.14 | **0.27** |
| Browsers | -0.09 | -0.13 | -0.01 | -0.11 |
| Video | **-0.28** | **-0.27** | **-0.28** | -0.21 |
| Writing | 0.24 | **0.27** | 0.10 | 0.09 |
| Search | -0.19 | -0.19 | -0.22 | -0.10 |
| Email | 0.21 | **0.28** | 0.11 | 0.10 |
| General News & Opinion | 0.19 | 0.04 | 0.14 | 0.13 |
| Games | 0.06 | -0.17 | 0.04 | -0.04 |
| Presentation | 0.14 | 0.10 | -0.01 | 0.03 |
| General Shopping | -0.05 | 0.02 | -0.12 | 0.02 |
| Music | 0.12 | 0.10 | 0.12 | 0.24 |

(Correlations in bold font are significant on a 5% level).

Satisfaction with life and mental well-being are positively correlated and are also significant predictors over time, i.e., subjective well-being in survey 1 is positively correlated with the same measure in survey 2. Regarding digital activities, we see, on average, negative correlations between subjective well-being and all digital activities, albeit not being significant. Similarly, we do not find significant correlations with social media use. (We will address causality in the section below.) To address potential non-linear effects we also report correlations with categories of social media usage. Specifically, we used dummy variables relating to low usage with an average of less than 2 minutes per day (36.1% of users), medium usage of 2 to 20 minutes (39.5%), and high usage of 20 minutes or more (24.4%). A non-linear relationship is likely as low usage generally shows the most negative subjective well-being scores, while medium usage and not high usage scores the highest well-being. However, we cannot rule out reverse causality regarding these findings as the only significant relationships are between satisfaction with life measured in survey 1 and the social media activities measured after the survey has taken place.

Activities related to communication, i.e., instant messaging and email, show significant positive correlations in the second surveys (for instant messaging regarding mental well-being and for email in terms of satisfaction with life). A consistent significant negative correlation can be observed for activities in the video category and satisfaction with life (both surveys) and mental well-being (survey 1). This suggests that less satisfied students and those with lower mental well-being at the beginning of the block increasingly watch videos in the subsequent block.

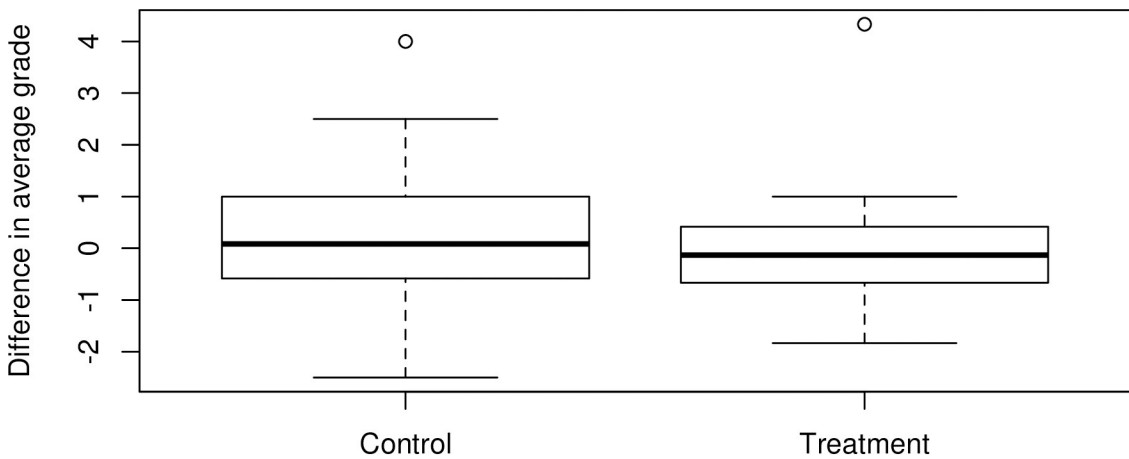

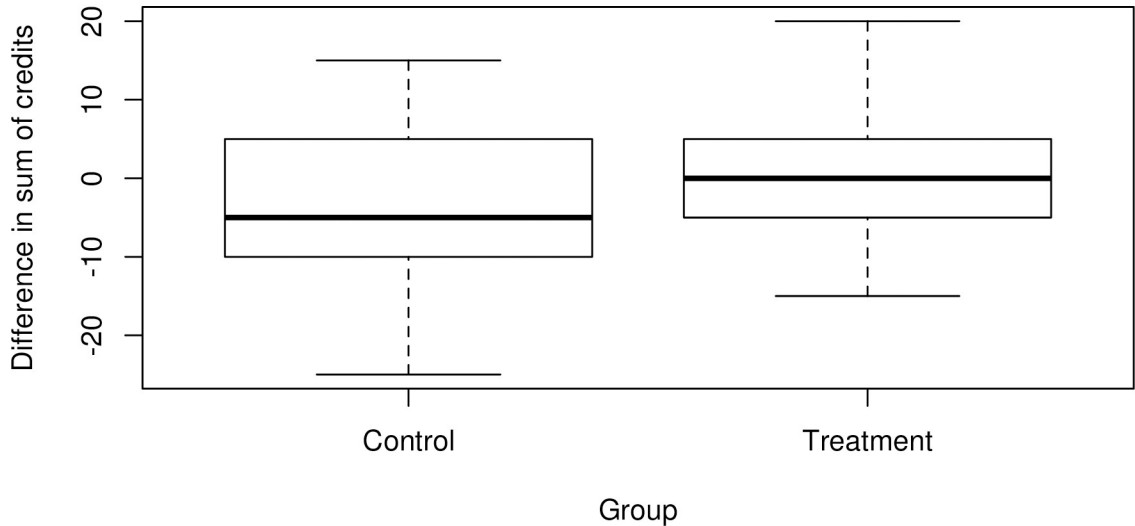

**Fig 4. Differences in academic performance.**

**Academic performance.** The students participating in our study scored an average (SD) of 7.105 (1.078) in block 1 and 7.122 (0.954) in block 2. Most grades can be classified as "standard". The average (SD) sum of credit points per block is 13.571 (5.754) in block 1 and 12.353 (5.118) in block 2. Differences between the treatment and control group are not significant for grades (t-test, p = 0.113) but for the number of credit points such that the treatment group attempted to score significantly more credits (t-test, p = 0.035). This is visualized in Fig 4 as the difference in grades and credits in block 1 and 2. Note that the number of ECTS represents the courses that the student *attempted* to pass but they are also stored if the student failed the exam. A comparison of the number of successfully passed courses shows no significant differences between the groups (t-test, p = 0.383).

**Table 5. Correlations of academic performance with measures of subjective well-being and digital activities in block 1 (Android users).**

|  | Average grade in block 1 | Sum of credit points in block 1 |
|---|---|---|
| Sum of credit points in block 1 | -0.16 | |
| Average grade in block 2 | **0.46** | -0.19 |
| Sum of credit points in block 2 | 0.06 | 0.01 |
| SWLS Survey 1 (beginning of block 1) | **0.27** | -0.10 |
| SWLS Survey 2 (end of block 1) | **0.31** | 0.02 |
| SWEMWBS Survey 1 (beginning of block 1) | 0.16 | -0.07 |
| SWEMWBS Survey 2 (end of block 1) | 0.22 | 0.01 |
| All digital activities | 0.16 | -0.16 |
| Social media usage | 0.02 | -0.03 |
| Social media usage low | -0.08 | 0.21 |
| Social media usage medium | 0.06 | **-0.26** |
| Social media usage high | 0.00 | 0.09 |
| General Reference & Learning | 0.13 | -0.21 |
| Instant Message | **0.28** | **-0.28** |
| Browsers | -0.03 | -0.18 |
| Video | -0.04 | 0.15 |
| Writing | **0.25** | 0.05 |
| Search | -0.13 | -0.21 |
| Email | 0.12 | -0.21 |
| General News & Opinion | -0.07 | -0.10 |
| Games | -0.06 | -0.08 |
| Presentation | **0.28** | 0.17 |
| General Shopping | -0.02 | -0.01 |
| Music | -0.08 | -0.13 |

(Correlations in bold font are significant on a 5% level).

Table 5 shows the correlations of academic performance in block 1 with the subjective well-being measures and digital activities. Accordingly, grades in block 1 are positively related to grades in block 2. The grades are not significantly correlated with the number of credit points, possibly due to a trade-off regarding a good grade and completing more courses. Credits are also not positively related over time, which is reasonable as more credits in one term means that the students have to obtain fewer credits in subsequent terms. The average grade is positively and significantly correlated with satisfaction with life measures. This holds for SWLS measures at the beginning and end of the block.

Regarding correlations with digital activities we can observe significant positive effects on the average grade for writing and presentation activities, which are required to complete assignments (that are part of the grade for the majority of courses). We also see a positive effect of instant messaging on the grade, however, a negative effect on the number of credit points. Social media usage is not significantly correlated with the academic performance in block 1. Only when categorizing students based on their social media usage we see significant effects such that medium usage is negatively correlated with the number of credits.

To what extent these findings can be interpreted as causal evidence will be addressed in the following section by analyzing the complete randomized control trial across the two blocks using difference-in-differences analyses with regression models.

**Table 6. Regression of satisfaction with life and mental well-being measures (standardized coefficients in parentheses).**

| | Satisfaction with life (SWLS) | | | | Mental well-being (SWEMWBS) | | | |
|---|---|---|---|---|---|---|---|---|
| | All users | | Android | | All users | | Android | |
| | Estimate | p-value | Estimate | p-value | Estimate | p-value | Estimate | p-value |
| (Intercept) | **29.028** | **<0.001** | **36.967** | **<0.001** | **24.409** | **<0.001** | **30.497** | **<0.001** |
| Treatment group | -1.205 (-0.113) | 0.221 | -1.350 (-0.117) | 0.380 | -0.233 (-0.038) | 0.676 | -0.063 (-0.009) | 0.942 |
| Block 2 | 0.207 (0.019) | 0.831 | -0.552 (-0.048) | 0.720 | 0.025 (0.004) | 0.963 | -0.342 (-0.050) | 0.691 |
| (Treatment group*Block2) | -0.253 (-0.020) | 0.855 | 1.285 (0.097) | 0.552 | -0.126 (-0.018) | 0.872 | 0.505 (0.065) | 0.676 |
| Gender (female) | -1.021 (-0.096) | 0.143 | -0.818 (-0.070) | 0.455 | **-0.991** **(-0.162)** | **0.013** | -0.768 (-0.112) | 0.210 |
| Age in years | -0.161 (-0.096) | 0.221 | -0.494 (-0.217) | 0.068 | -0.057 (-0.059) | 0.446 | **-0.338** **(-0.252)** | **0.026** |
| Years at the university | 0.166 (0.048) | 0.542 | 0.107 (0.028) | 0.798 | -0.139 (-0.069) | 0.370 | -0.089 (-0.040) | 0.705 |
| Working next to studies | 0.280 (0.026) | 0.692 | -0.036 (-0.003) | 0.975 | 0.724 (0.117) | 0.072 | 1.020 (0.150) | 0.117 |
| R-squared | 0.032 | | 0.051 | | 0.055 | | 0.142 | |

**Regressions.** Tables 6 and 7 show the results of the regression models that use subjective well-being (Table 6) and academic performance measures (Table 7) as the dependent variables. We use data from the two teaching blocks with "block 2" being a dummy variable indicating the block in which the treatment took place. Similarly, the "treatment group" refers to a dummy variable that identifies students that were exposed to the treatment. We use gender, age, number of years at the university, and whether the student is working next to the studies (dummy variable) as control variables.

**Table 7. Regression of academic performance (standardized coefficients in parentheses).**

| | Regression of average grade | | | | Regression of number of credit points | | | | Regression of number of courses passed | | | |
|---|---|---|---|---|---|---|---|---|---|---|---|---|
| | All users | | Android users | | All users | | Android users | | All users | | Android users | |
| | Estimate | p-value | Estimate | p-value | Estimate | p-value | Estimate | p-value | Estimate | p-value | Estimate | p-value |
| (Intercept) | **6.950** | **<0.001** | **6.578** | **<0.001** | **19.406** | **<0.001** | **13.393** | **0.009** | **3.904** | **<0.001** | **2.616** | **0.007** |
| Treatment group | 0.237 (0.117) | 0.205 | 0.536 (0.244) | 0.067 | -1.740 (-0.159) | 0.071 | -1.159 (-0.108) | 0.388 | -0.064 (-0.032) | 0.721 | 0.124 (0.064) | 0.625 |
| Block 2 | 0.173 (0.086) | 0.350 | 0.321 (0.146) | 0.272 | **-2.750** **(-0.252)** | **0.004** | **-2.931** **(-0.273)** | **0.031** | -0.347 (-0.172) | 0.051 | -0.345 (-0.177) | 0.178 |
| (Treatment group*Block2) | -0.310 (-0.132) | 0.239 | -0.592 (-0.235) | 0.149 | **3.090** **(0.245)** | **0.023** | 3.431 (0.278) | 0.070 | 0.220 (0.094) | 0.382 | 0.211 (0.094) | 0.554 |
| Gender (female) | 0.142 (0.070) | 0.281 | 0.137 (0.062) | 0.507 | 0.706 (0.065) | 0.300 | 0.388 (0.036) | 0.684 | 0.201 (0.099) | 0.113 | 0.090 (0.046) | 0.618 |
| Age in years | -0.016 (-0.048) | 0.534 | 0.002 (0.005) | 0.969 | -0.130 (-0.076) | 0.311 | 0.164 (0.078) | 0.483 | **-0.055** **(-0.172)** | **0.022** | 0.003 (0.007) | 0.949 |
| Years at the university | **0.114** **(0.172)** | **0.028** | 0.083 (0.116) | 0.294 | **-0.854** **(-0.239)** | **0.002** | **-1.232** **(-0.352)** | **0.001** | **-0.109** **(-0.165)** | **0.028** | -0.131 (-0.206) | 0.060 |
| Working next to studies | -0.043 (-0.021) | 0.750 | -0.202 (-0.092) | 0.358 | -0.009 (-0.001) | 0.990 | 1.439 (0.134) | 0.156 | -0.082 (-0.040) | 0.523 | 0.134 (0.069) | 0.484 |
| R-squared | 0.034 | | 0.054 | | 0.117 | | 0.159 | | 0.113 | | 0.083 | |

Regarding subjective well-being measures SWLS and SWEMWBS, we do not see any significant effects due to using Facebook, Instagram, and Snapchat less. The (Treatment group * Block2 interaction) that indicates if the treatment group differed from the control group in block 2 remains insignificant, irrespectively of the well-being measure (SWLS or SWEMWBS) or sample used (full sample or Android subset). Only few predictors are significant (gender in the full sample, p = 0.013 or age in the Android subset, p = 0.026; both for the SWEMWBS) but these are not directly related to the experimental setting. Overall, the amount of variance explained remains rather low and ranges between 3.2% (SWLS measure, full sample) and 14.2% (SWEMWBS, Android sample).

There is also no evidence that the treatment group achieves higher grades in block 2 (Treatment group * Block2 interaction), in which social media usage was restricted, than the control group (p = 0.239). This also holds for other subsets of the sample such as students with an Android device. Overall, the amount of variance explained in the grades remains very low with 3.4%. Only the number of years that the student has spent at the university is a significant positive predictor, i.e., Master's students achieve higher grades than Bachelor's.

In terms of number of credit points, i.e., number of courses attended, we do see a significant effect of restricting social media. While students overall attended courses with fewer credits in block 2 (p = 0.004) this is not the case for students in the treatment group such that they targeted significantly more credits (p = 0.023). With this dependent variable, the number of years at the university has a significant negative effect (p = 0.002) and, overall, 11.7% of variance in the number of credit points can be explained. The subset of Android users replicates these results, albeit generally with lower levels of significance due to the smaller sample size.

However, as noted above, the number of credit points does not necessarily show that students successfully passed more courses as also failed courses are included. Using the number of courses passed as the dependent variable shows that the students in the treatment group in fact do not differ from the control group (a Poisson model replicates these results). Thus, it appears that the treatment group attempted to pass more courses or courses with more credits compared to the control group but did not necessarily succeed.

These null results of the effects of restricting social media usage on academic performance and subjective well-being raises the question of whether our study is underpowered to detect economically significant effects. At 80% power the minimum detectable difference in life satisfaction (on the SWLS scale) with our sample size is 2.5 on a scale of 5–35 (average life satisfaction score in our sample is 25 with a standard deviation of 5). The SWLS categories comprise 5 scale points, for example, a score of 25–29 is considered as a "high" score. Therefore, even if the treatment group's life satisfaction score increased by 1.5 it is generally not sufficient to change the classification from one category to another. The minimum detectable difference in average grade is 0.5 on a scale of 1–10 (average grade in our sample is about 7 with a standard deviation of 1). Given that students receive grades which are whole numbers, this threshold is still below the value that would increase the treatment group's average grade by a full point. At least, our results indicate that changes in grades and subjective well-being are not substantial so that they could be detected with our sample size.

To further address potential concerns about statistical power we applied a hierarchical Bayes ANOVA (BANOVA) model that includes between and within subject effects and accommodates unobserved heterogeneity by including a normal distribution of the parameters across individuals [51]. All models converge and generally replicate the results above (details are available from the authors upon request).

**Post-treatment effects.** The formal analysis of the post-treatment effects is based on a sample of 106 students who provided activity data throughout all three blocks and in all four surveys. While the treatment condition significantly reduced their social media usage in block

2 compared to the control group, this effect was not permanent. After we suspended the limit in block 3 the social media activities of users in the treatment group increased again, showing no significant differences to the control group any longer (t-test, p = 0.668). We further do not see any significant differences between the treatment and control group in block 3 in terms of grades (t-test, p = 0.152), number of credit points (t-test, p = 0.923), satisfaction with life (t-test, p = 0.499), or mental well-being (t-test, p = 0.966), i.e., there is no lagged effect of reduced social media usage.

## Discussion

In this paper, we analyzed the effects of restricting social media usage. We did not find significant causal effects of social media usage on well-being or academic performance, other than students attempting (but not succeeding) to pass more courses or courses with more credits. However, we found robust evidence of substitution effects that can potentially explain the null finding. Specifically, we showed that social media and instant messaging apps can be substitutes. The European Commission approved Facebook's acquisition of WhatsApp in 2014 based on Facebook's claim that it operates in a different market and does not compete directly with WhatsApp [52]. Our results indicate that they are in fact direct competitors. After acquiring WhatsApp, Facebook started automatically matching its users' profiles with their WhatsApp accounts and has started integrating WhatsApp, Instagram and Facebook user accounts [53]. The European Commission fined Facebook €110 million in 2017 for this practice because Facebook had provided misleading information about the feasibility of automatically matching profiles during its acquisition of WhatsApp. However while announcing this fine, the Commission still maintained its belief that Facebook and WhatsApp do not directly compete with each other [54]. Antitrust authorities should consider the market power of this combined entity if the world's biggest social media platforms are integrated with the world's biggest instant messaging platform. This finding also raises the question how to properly define social networks, for future academic studies or antitrust cases.

While we found null results estimating the causal impact of social media usage on well-being and academic performance, and not all null results matter, we believe that null results are interesting and important in this context. The media has hyped correlational studies showing a negative association between social media usage and well-being and it is important to balance this narrative through causal evidence. A limitation of our study is the lack of a larger sample size to detect smaller effects. While these small effects might not be economically significant, more research is needed using massive samples. Future research can also look at differences between Android and iOS users in more detail with larger samples since we lack power to analyze these differences in our current study. However, it is challenging to recruit a large number of subjects from a representative sample for a long-term study. Direct collaborations with social media platforms or internet service providers (which control internet traffic) could be a way of obtaining data from larger samples. These would also facilitate more targeted interventions such as restricting only content consumption or social interactions on social media [18].

It is interesting to notice that while social media generates large amount of consumer surplus [3], it doesn't seem to affect the subjective well-being of users. Future research can explore this wedge between consumer surplus and subjective well-being and see whether they are correlated for some products and uncorrelated for others. Future research should also explore the addictiveness of social media in more detail [55]. Our findings in block 3 show that the students in the treatment condition go back to their old habits and do not adopt a lower social media usage that they experienced in block 2. On the other hand, showing students how much

time they are spending on social networks via the software seems to have an overall negative trend on its usage (comparing usage in block 1 and block 3). Curing social media addiction (if it is indeed addiction) might therefore be a longer process. Future research can look at mechanisms for the emergence of social media addiction, for e.g. through targeted advertising or news feed algorithms and features of social media apps (e.g. video sharing) which are correlated with addiction.

Moreover, due to our student sample implications for the general population are limited. It could be that students use social media mostly for communication purposes and therefore show significant substitution effects with instant messaging. We might see different effects for users who visit social media for content consumption, e.g., watching videos, instead of social interaction [18]. In this regard, newer social media services such as TikTok might have a different effect. However, these and other comparable social media services were not relevant at the time we conducted the study (as we measured in block 1). More research is needed looking at emerging social media platforms such as TikTok. Moreover, one could define social media more broadly to include instant messaging services and implement a stricter restriction involving all types of communication and social networking apps. However, conducting such a study for any length of time is a challenging endeavor.

Additionally, while we study the impact of social media on students and academic performance, future research can look at workplace settings and study the impact of social media and its substitutes on worker productivity and well-being. We believe that rigorous causal evidence through randomized controlled trials and objectively measured time spent is the way forward in addressing questions regarding the impact of technology on well-being.

The widespread adoption of most major technologies in the past such as radio, television, video games and computers was followed with unfounded fears about their impact on well-being. This story repeats again with social media. We find that social media usage does not cause lower well-being or poor academic performance. Rather, we demonstrate that students find other means of social networking using instant messaging when exogenously restricting their social media usage. To conclude: You can take social networking away from the students, but you cannot take students away from their social network.

## Supporting information

**S1 Fig. All digital activities and social media usage over time (all users).**
(DOCX)

**S2 Fig.** a Tracked digital activities over time (all users). b. Tracked digital activities over time (all users).
(DOCX)

**S1 Table. Significance of differences between treatment and control group in block 2 vs. block 1 (difference-in-differences).**
(DOCX)

**S2 Table. Summary statistics for well-being measures.**
(DOCX)

## Author Contributions

**Conceptualization:** Avinash Collis, Felix Eggers.

**Data curation:** Avinash Collis, Felix Eggers.

**Formal analysis:** Avinash Collis, Felix Eggers.

**Funding acquisition:** Avinash Collis, Felix Eggers.

**Investigation:** Avinash Collis, Felix Eggers.

**Methodology:** Avinash Collis, Felix Eggers.

**Project administration:** Avinash Collis, Felix Eggers.

**Resources:** Avinash Collis, Felix Eggers.

**Software:** Avinash Collis, Felix Eggers.

**Supervision:** Avinash Collis, Felix Eggers.

**Validation:** Avinash Collis, Felix Eggers.

**Visualization:** Avinash Collis, Felix Eggers.

**Writing – original draft:** Avinash Collis, Felix Eggers.

**Writing – review & editing:** Avinash Collis, Felix Eggers.

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
