## [Decision Letter · Decision Letter 0]

20 Aug 2021

PONE-D-21-20386

Effects of restricting social media usage

PLOS ONE

Dear Dr. Collis,

Thank you for submitting your manuscript to PLOS ONE. After careful consideration, we feel that it has merit but does not fully meet PLOS ONE’s publication criteria as it currently stands. Therefore, we invite you to submit a revised version of the manuscript that addresses the points raised during the review process.

My summary of the major concerns raised by the reviewers is below and will hopefully give you guidance in how to respond if you feel that you can.

We look forward to receiving your revised manuscript.

Kind regards,

Daniel Romer

Academic Editor

PLOS ONE

Journal Requirements:

2. Please consider changing the title so as to meet our title format requirement (https://journals.plos.org/plosone/s/submission-guidelines). In particular, the title should be "Specific, descriptive, concise, and comprehensible to readers outside the field" and in this case it is not informative and specific about your study's scope and methodology.

Additional Editor Comments

We have received two reviews that I think will be helpful to you if you wish to pursue publication. In particular, both reviewers think your study is under-powered considering the typical effect size that is observed in the literature. This may be a serious obstacle to moving ahead with the study as it stands. However, R2 suggests that perhaps a multi-level analysis may increase your power and thus overcome this problem,. R1 also suggests that it may be valuable to examine effects in terms of mediators that have produced different effects and thus hindered your ability to observe an effect (i.e., the type of content that is accessed). If this objection can be overcome, then it may be possible to salvage your study. In addition, both reviewers think that your study is limited in so far as it only required reductions in use of three platforms. Can you respond to this concern by looking at what the students actually did in each condition? Obviously, if the experimental group merely displaced use to other platforms, then the findings are much more limited in terms of overall social media use. In addition, from my own perspective, I would look to see if any effects are more likely among those who were already experiencing mental health problems because withdrawal from in-person interaction is common among those with depression and related conditions. it may be that they are the ones who are likely to show stronger effects of not using the one socializing outlet they find attractive. Finally, R1 asks for more disclosure of findings that would be helpful to others who would want to include your findings in a synthesis of results, and both reviewers feel that important prior papers are being ignored in your literature review.

Reviewers' comments:

Reviewer's Responses to Questions

**Comments to the Author**

1. Is the manuscript technically sound, and do the data support the conclusions?

Reviewer #1: Partly

Reviewer #2: Partly

2. Has the statistical analysis been performed appropriately and rigorously? 

Reviewer #1: No

Reviewer #2: Yes

3. Have the authors made all data underlying the findings in their manuscript fully available?

Reviewer #1: No

Reviewer #2: No

4. Is the manuscript presented in an intelligible fashion and written in standard English?

Reviewer #1: Yes

Reviewer #2: Yes

5. Review Comments to the Author

Reviewer #1: Review for PONE-D-21-20386 : Effects of restricting social media usage

This paper describes a random assignment experiment comparing the subjective well-being (satisfaction with life and mood) and grades of university students who were asked to reduce their social media use (Facebook, Instagram, Snapchat) to less than 10 minutes per day with a control group who were not asked to make social media adjustments. Results show no significant differences between the reduction group and control group on these outcomes. (Because I am not familiar with work that on social media use and educational outcomes, I will ignore it in this review.)

As the authors note, prior research on the impact of social media use on subjective well-being has primarily involved cross-sectional surveys, correlating self-reports of social media use with self-reports of well-being. Moreover, again as the authors note, results from the correlational research have been mixed, although the most recent meta-analysis reviewing 94 studies (Liu et al., 2019) suggests that on average social media use is negatively associated with subjective well-being. However, the association is substantially dependent on type of use (e.g., r=.14 for social interaction versus -.14 for content consumption). Thus, a random assignment experiment is an excellent addition to this body of work.

The authors introduce their research with the claim for novelty. “To the best of our knowledge, this study is the first that tracks all of these components of well-being and over a long period of time.” This claim is overstated. I am familiar with five experiments, in which participants were randomly assigned to reduce their social media use for one to four weeks (Allcott et al., 2020; Brailovskaia et al., 2020; Hanley et al., 2019; Mosquera et al., 2019; Vanman et al., 2018). The current research improves on prior experiments by upping the ante to nine weeks. Although some of the prior experiments used small student samples like the current study, others used much larger and more diverse samples. All of them show that social media restrictions led to better subjective well-being. Of these, the current authors cite only the Allcott paper, but criticize it because it only reduced social media use for a four-week period.

I have some substantive methodological concerns about the paper involving the conceptualization of the intervention, statistical methods, and the power of the intervention. Given the prior literature, which suggests that the impact of social media use depends upon the functions for which it is used and not just the total amount of use, I am disappointed that the current research did not target more precisely the type of social media use to restrict. For example, a 2 X 2 factorial experiment (reduce social interaction vs control (reduce content consumption vs control) would have been more informative than the current experiment, which restricted overall use, and could have been done even with a relatively small sample size of 130 completed participants.

I also disappointed on the way that the research defined social media use for participants. The research considered Facebook messenger (a popular application on Facebook), Instagram and Snapchat, all of which are used for one-on-one or small group communication, as social media applications while WhatsApp, another messaging application, is not classified as social media.

According to the methods description on page 13 and Tables 4 and 5 in the manuscript, the authors have some data on what activities participants used across the platforms. I think the authors could more strongly advance the prior literature about the differential importance of types of use and still take advantage of their random assignment experiment by using instrumental variable regression or mediation models to analyze the data. For example, the authors could use the experiment to dig more deeply into the conclusion form Liu et al’s meta-analysis that social interaction and content consumption have different effects on well-being. An instrumental variable regression analysis could examine the extent to which the experimental manipulation changed participants’ type of use (e.g., instant messaging and email as proxies for use for social interaction vs general news, general shopping, and video as proxies for content consumption) and how the changes in in use for social interaction and content consumption use caused by the manipulation predict well-being and grades.

I have two concerns about the power of the experiment. First, the manipulation check indicates that the experimental manipulation did not have a major impact on the amount of time participants spent on social media, with participants in the experimental condition reducing time spent on the three predefined social media applications by 13 minutes versus 3.7 minutes for participants in the control condition. Given the substitution behavior the authors document, in which participants in experimental condition started using WhatsApp more, the experimental manipulation did not reduce overall social media use. Instead, it caused shifts in the types of social media applications used, if one considers Facebook, Instagram, Snapchat, and WhatsApp all as social media platforms. This shift in behavior is another reason I recommend that the authors consider an instrumental variable regression model to test their hypotheses.

Second, both the correlational and experimental prior literature suggests that the relationship between social media use and well-being is likely to be small, with the average absolute correlation being -.10 in the Liu meta-analysis of correlational studies and a .09 standard deviation (STD) effect size in the Allcott et al experiment. Given the likely small effect size, an experiment with 130 participants is probably underpowered to detect this effect. The authors’ discussion of their power analysis indicates that their study could detect a .6 STD effect on the SWLS Life Satisfaction scale, substantially larger than the effect size one might have anticipated from the prior literature. While the authors acknowledge this limitation, they still interpret their failure to find a significant effect of their restriction intervention much more seriously than it deserves. As their paper states, “Contrary to findings from previous correlational studies, we do not find any impact of social media usage on wellbeing and academic success. … While we found null results estimating the causal impact of social media usage on well-being and academic performance, and not all null results matter, we believe that null results are interesting and important in this context. The media has hyped correlational studies showing a negative association between social media usage and well-being and it is important to balance this narrative through causal evidence.” Indeed, even though the experiment found no significant effects of their intervention on either satisfaction with life or mental well-being, the treatment X Block2 effects are both negative and consistent with the prior literature. As part of this review, I had intended to conduct a quick and dirty meta-analysis of the experimental literature to date (i.e., the current paper plus the five additional experiments listed below) but discovered that the authors of the current paper don’t provide enough information to include their research in the meta-analysis. Page 17, where the authors present their well-being results, should include a table of means and standard deviations for the SWLS and SWEMWBS scores for Block 2 broken down by experimental condition. Table A-3, which only reports the regression coefficients and p-values does not provide sufficient information to reconstruct the needed information.

In summary, this is an interesting study with a laudable goal. Experimental evidence is hugely important, and the current study supplements other recent experiments restricting social media usage by throttling back usage over a 9-week period, compared to a maximum of four weeks in other work. However, the study has some important flaws that make me believe the paper shouldn’t be published in its current form. The authors made arbitrary decisions about what online behavior to consider use of social media (i.e., only Facebook, Snapchat, and Instagram, but not WhatsApp), failed to differentiate types of use (e.g., social interaction versus content consumption) even though the prior literature suggest that type of use is an important moderator, and was under powered. A more minor problem is that the authors don’t provide enough information to allow their research to be used in a meta-analysis. I think the paper still has promise but needs to do more sophisticated data analysis (e.g., Instrumental variable approach mentioned previous or a mediation analysis) to link the changes induced by the experimental manipulation on well-being.

References

Allcott, H., Braghieri, L., Eichmeyer, S., & Gentzkow, M. (2020). The welfare effects of social media. American Economic Review, 110(3), 629-676.

Brailovskaia, J., Ströse, F., Schillack, H., & Margraf, J. (2020). Less Facebook use–More well-being and a healthier lifestyle? An experimental intervention study. Computers in Human Behavior, 108, 106332.

Hanley, S. M., Watt, S. E., & Coventry, W. (2019). Taking a break: The effect of taking a vacation from Facebook and Instagram on subjective well-being. PloS one, 14(6), e0217743.

Liu, D., Baumeister, R. F., Yang, C.-c., & Hu, B. (2019). Digital communication media use and psychological well-being: A meta-analysis. Journal of Computer-Mediated Communication, 24(5), 259-273.

Mosquera, R., Odunowo, M., McNamara, T., Guo, X., & Petrie, R. (2019). The economic effects of Facebook. Experimental Economics, 1-28.

Vanman, E. J., Baker, R., & Tobin, S. J. (2018). The burden of online friends: The effects of giving up Facebook on stress and well-being. The Journal of social psychology, 158(4), 496-508.

Reviewer #2: The paper submitted to PLOS One investigates whether voluntary restriction of social media usage to 10 minutes per day has any effects on a) subjective well-being and b) academic performance. To this end, the authors conducted an randomised controlled trial in which half of 121 students were asked to restrict their social media usage (i.e., the use of social network sites Facebook, Instagram and Snapchat) to a maximum of 10 minutes per day over a period of 1.5 months. Overall, the authors did not find any effects of social media usage restriction on subjective well-being or academic performance.

First, I would like to highlight that I deem the topic – social media uses effects on well-being/academic performance – and the proposed experimental approach to study it – experimentally restricting social media use – highly important and innovative. The debate around effects of social media use or screen time on well-being is on-going, both in public and scientific circles. And the authors are right to point out that much of it is based on questionable, correlational evidence. From this point of view, I welcome the submission. Second, the experiment is well designed and conducted, the analyses seem for the most part sound, and the paper overall well-written. The author neither make to strong claims, nor fail to articulate the contribution. As such, I do believe that the paper has the potential to add to the literature.

However, I do see several problems and issues, most minor but some also major, that require revision.

- First and foremost, I am concerned that the authors did not conduct comprehensive a priori power analyses. The sample sizes after each block are comparatively small and hardly sufficient to test effects of reduced social media use on general and stable outcomes such as well-being or academic performance. In other words, the study seems heavily underpowered to study the effects of interest. I believe that this is a major issues that unfortunately might not be resolvable. I thus recommend the authors to consider conducting another study with a more appropriate sample size. When doing so, I would also recommend to preregister all assumptions and analysis plans.

- Second, although the authors provide a summary of related literature, it – from my point of view – does not reflect the current state of the debate sufficiently. A lot of major contributions in the past years are missing and some similar experiments are not mentioned (among others, I was particularly missing reference to the work of Jeffrey Hall, 2021).

- Another issue relates to a lack of transparency in reporting. Some important aspects of the data analysis are not reported (e.g., missing data analysis and treatment, exact tests and covariates...). Information about the used tracking app (including how well the capturing of data worked) is missing completely. I appreciate that the authors explain why the data cannot be shared, but I strongly recommend to make everything else (material, instructions, information about the app, analysis code, etc.) available to the reader.

In the following, I will highlight more specific issues by going through the manuscript chronologically.

pp. 3-9.: As mentioned above, I am missing a comprehensive discussion of the current state of the literature. With regard the relationship between social media use and well-being, consider referring to the meta-analyses Huang (2010, 2017), Liu, Baumeister, Yang & Hu (2019), as well as the reviews by Appel, Marker & Gnambs (2020), Meier & Reinecke (2021), Dienlin & Johannes (2020), Masur (2021) and Orben (2020). Consider also looking at more recent studies on the relationship, e.g., Schemer et al. (2020), Kim & Chen (2020), Johannes et al. (2020). With regard to experimental studies on effect of social media abstinence, I was further missing reference to the work by Jeffrey Hall (2021) and the earlier works by Hinsch & Sheldon (2013) and Tromholt (2016). I strongly recommend that the authors revise this sections considerably.

p. 9: Consider explicitly formulating your hypotheses. In its current form, I find it difficult to understand what you are exactly testing.

p. 10: What was the exact reason to assess participants at 3 different time points? And why did you not incorporate this in your analysis? Wouldn't it be possible to run some sort of time-series or multi-level analysis on the entire data set? This way, you could potentially also increase your power (by combining within- and between-perspectives).

p. 11: The paper is framed as "reducing social media use". The term social media is quite broad. Often, it even includes instant messaging and other types of platforms. In light of this, why only restricting Facebook, Instagram and Snapchat use? What about TikTok, Twitter, Pinterest, Youtube? I suggest to include clear definition of social media and justify your choice of platforms. This may have important implications for how to frame the entire study and how to intrepret the results as well: I would be careful to generalize on all social media use given that the treatment was restricted to three platforms (albeit the most used ones).

p. 12: Although perhaps not very common, I would appreciate you reporting comprehensive factor analyses (CFAs, etc) for both well-being scales. Despite them being used a lot, they often show considerable misfit with the data.

p. 13: What type of software/application was used? Please provide more information and validity checks. Did the tracking work perfectly? From my experience, one often has some problems in the data with such software.

p. 14: What was your justification for the sample size? As mentioned earlier, I am concerned that your study is heavily underpowered. Further, how did you deal with missing data? Did all 122 participants answers all items in all waves?

p. 15: In reporting the results, it is often unclear what tests of analytical approaches were chosen. Please indicate more clearly which results relate to which analysis. Are the p-values related to simple t-tests? Or more elaborate ANOVAS including covariates? This should be made more clearer. Please report unstandarized and standardized coefficients (effect sizes) as well. Significance alone is not meaningful.

References

Appel, M., Marker, C., & Gnambs, T. (2020). Are social media ruining our lives? A review of meta- analytic evidence<https: 10.1177="" 1089268019880891="" doi.org="">. Review of General Psychology, 24(1), 60–74.

Dienlin, T., & Johannes, N. (2020). The impact of digital technology use on adolescent well- being<https: 10.31887="" dcns.2020.22.2="" doi.org="" tdienlin="">. Dialogues in Clinical Neuroscience, 22(2), 135–142.

Hall, J. A., Xing, C., Ross, E. M. & Johnson, R. M. (2021) Experimentally manipulating social media abstinence: results of a four-week diary study.Media Psychology, 24:2, 259-275, DOI: 10.1080/15213269.2019.1688171

Hinsch, C., & Sheldon, K. M. (2013). The impact of frequent social internet consumption: Increased procrastination and lower life satisfaction. Journal of Consumer Behaviour, 12, 496–505. doi:10.1002/cb.1453 

Huang, C. (2010). Internet use and psychological well-being: A meta-analysis. Cyberpsychology, Behavior, and Social Networking, 13(3), 241–249.

Huang, C. (2017). Time spent on social network sites and psychological well-being: A meta- analysis<https: 10.1089="" cyber.2016.0758="" doi.org="">. Cyberpsychology, Behavior, and Social Networking, 20(6), 346–354.

Johannes, N., Meier, A., Reinecke, L., Ehlert, S., Setiawan, D. N., Walasek, N., Dienlin, T., Buijzen, M., & Veling, H. (2020). The relationship between online vigilance and affective well-being in everyday life: Combining smartphone logging with experience sampling<https: 10.1080="" 15213269.2020.1768122="" doi.org="">. Media Psychology.

Kim, C., & Shen, C. (2020). Connecting activities on social network sites and life satisfaction: A comparison of older and younger users<https: 10.1016="" doi.org="" j.chb.2019.106222="">. Computers in Human Behavior, 105, 106222.

Liu, D., Baumeister, R. F., Yang, C., & Hu, B. (2019). Digital communication media use and psychological well-being: A meta-analysis<https: 10.1093="" doi.org="" jcmc="" zmz013="">. Journal of Computer-Mediated Communication, 24(5), 259–273.

Masur, P. K. (2021). Digital Communication Effects on Loneliness and Life Satisfaction. In J. Nussbaum (Ed.), Oxford Research Encyclopedia of Communication. Oxford University Press. https://doi.org/10.1093/acrefore/9780190228613.013.1129

Meier, A., & Reinecke, L. (2020, October 21). Computer-mediated communication, social media, and mental health: A conceptual and empirical meta-review<https: 0093650220958224="" 10.1177="" doi.org="">. Communication Research.

Orben, A. (2020b). Teenagers, screens and social media: A narrative review of reviews and key studies<https: 10.1007="" doi.org="" s00127-019-01825-4="">. Social Psychiatry and Psychiatric Epidemiology, 55(4), 407–414.

Schemer, C., Masur, P. K., Geiss, S., Müller, P., & Schäfer, S. (2020). The impact of internet and social media use on well-being: A longitudinal analysis of adolescents across nine years<https: 10.1093="" doi.org="" jcmc="" zmaa014="">. Journal of Computer-Mediated Communication, 26(1), 1–21.

Tromholt, M. (2016). The Facebook experiment: Quitting Facebook leads to higher levels of well-being. Cyberpsychology, Behavior, and Social Networking, 19(11), 661–666. doi:10.1089/cyber.2016.0259</https:></https:></https:></https:></https:></https:></https:></https:></https:>

6. PLOS authors have the option to publish the peer review history of their article (what does this mean?). If published, this will include your full peer review and any attached files.

Reviewer #1: No

Reviewer #2: No

---

## [Author Response · Author response to Decision Letter 0]

14 Mar 2022

We uploaded a document responding point by point to each editor and reviewer comment.

---

## [Editor Report · Decision Letter 1]

13 Apr 2022

PONE-D-21-20386R1Effects of restricting social media usage on wellbeing and performance: A randomized control trial among studentsPLOS ONE

Dear Dr. Collis,

Thank you for making the extensive changes to your paper in response to the reviewers’ and my suggestions. The paper is much stronger but I have a few more suggestions before moving on to making a decision on publication.

First, a very real concern is the statistical power of your study, and I don’t think your description of that problem on page 27 is very clear or helpful. First of all, we don’t usually consider power from the point of view of the maximum difference we cannot detect. It would be easier to understand this if you talked about the minimum difference you can detect. Second, I am puzzled as to why you are talking about increases in life satisfaction, when the concern is for decreases. Also, should you not be citing the differences you observed and discussing whether you could have found those to be significant with more power? That is what seems to be the issue. Given that you found a slight increase among the Android group, it might make sense to look at both groups separately in considering the power question. I would also recommend discussing power in the method section with how large a difference you will be able to detect given the sample size you have. Then in the Results, you could comment on whether your differences are large enough to have rejected the null hypothesis with more power.

I am also puzzled as to why you show the regressions for grades but not for wellbeing and life satisfaction in the Results. It seems to me that the mental health concerns are the ones that motivated this to begin with, so those findings should receive priority.

Finally, I think your findings could use more discussion than you devote to them. It seems that your intervention to reduce some forms of social media use served to increase other forms of interaction within one’s social network. So, what your findings seem to say is that social media may primarily serve as a way to connect with others and that arbitrarily defining those media as you have merely caused your students to migrate to other platforms that serve the same purpose. In other words, what your study shows is that with all of the ways that people can interact online, it is difficult to isolate some as more responsible for adverse effects. You would have to restrict the ability to use online platforms to communicate with others as a way to isolate that effect and I doubt that you could recruit a sample to do that for any length of time.

One other possible way that some social media may be harmful is by pushing advertising to users that they may not want exposure to and doing this to maintain continued engagement.  This might be a mechanism for internet addiction that could be studied. In any case, I think a deeper discussion of these issues would be helpful for readers. Your point about the policy implications also follows from this, since many online media serve the same purpose and just differ in how they afford that opportunity.

I also think you have more findings to discuss, such as correlations between mental health and various uses of social and other media, such as in Table 4. Perhaps it is the video features of online media that are the source of adverse effects on mental health? And perhaps the sites push certain videos in order to maintain engagement?

In the abstract, I would describe the finding by saying that “we do not find any significant impact of social media usage as it was defined in this study on well-being…” As part of your limitations section, I would also take note of the fact that other forms of social media were not excluded in your study and this will be an obvious point of criticism.

I also think it would be useful to consider a paper that just came in Clinical Psych Science by Sewall et al. They also find that various uses of digital tech are not very strongly related to mental health in young adults

We look forward to receiving your revised manuscript.

Kind regards,

Daniel Romer

Academic Editor

PLOS ONE
---

## [Author Response · Author response to Decision Letter 1]

28 Jun 2022

Please see the attached document.

---

## [Editor Report · Decision Letter 2]

20 Jul 2022

Effects of restricting social media usage on wellbeing and performance: A randomized control trial among students

PONE-D-21-20386R2

Dear Dr. Collis,

We’re pleased to inform you that your manuscript has been judged scientifically suitable for publication and will be formally accepted for publication once it meets all outstanding technical requirements.

Kind regards,

Daniel Romer

Academic Editor

PLOS ONE

Additional Editor Comments (optional):

I found two typos and there may be more. i am attaching the pdf with those places highlighted.
---

## [Editor Report · Acceptance letter]

3 Aug 2022

PONE-D-21-20386R2 

Effects of restricting social media usage on wellbeing and performance: A randomized control trial among students 

Dear Dr. Collis:

I'm pleased to inform you that your manuscript has been deemed suitable for publication in PLOS ONE. Congratulations! Your manuscript is now with our production department. 

Kind regards, 

on behalf of

Dr. Daniel Romer 

Academic Editor

PLOS ONE